# *Listeria monocytogenes*: An Inconvenient Hurdle for the Dairy Industry

**Alessandra Casagrande Ribeiro** [1,†], **Felipe Alves de Almeida** [2,†], **Mariana Medina Medeiros** [1],
**Bruna Ribeiro Miranda** [3], **Uelinton Manoel Pinto** [1] and **Virgínia Farias Alves** [3,*]

1   Food Research Center (FoRC), Departamento de Alimentos e Nutrição Experimental, Faculdade de Ciências Farmacêuticas, Universidade de São Paulo (USP), São Paulo 05508-080, Brazil; alessandracrib@usp.br (A.C.R.); mariana.medeiros@usp.br (M.M.M.); uelintonpinto@usp.br (U.M.P.)
2   Instituto de Laticínios Cândido Tostes (ILCT), Empresa de Pesquisa Agropecuária de Minas Gerais (EPAMIG), Juiz de Fora 36045-560, Brazil; felipe.almeida@epamig.br
3   Faculdade de Farmácia, Universidade Federal de Goiás (UFG), Goiânia 74605-170, Brazil; bruna.ribeiro@discente.ufg.br
*   Correspondence: virginia_alves@ufg.br
†   These authors contributed equally to this work.

**Abstract:** *Listeria monocytogenes* is an opportunistic pathogen that affects specific groups of individuals, with a high mortality rate. The control of *L. monocytogenes* in dairy industries presents particular challenges, as this bacterium is capable of adhering and forming biofilms, as well as thriving at refrigerated temperatures, which enables it to persist in harsh environments. The consumption of dairy products has been linked to sporadic cases and outbreaks of listeriosis, and *L. monocytogenes* is frequently detected in these products in retail stores. Moreover, the bacterium has been shown to persist in dairy-processing environments. In this work, we review the main characteristics of *L. monocytogenes* and listeriosis, and highlight the factors that support its persistence in processing environments and dairy products. We also discuss the main dairy products involved in outbreaks of listeriosis since the early 1980s, and present control measures that can help to prevent the occurrence of this pathogen in foods and food-processing environments.

**Keywords:** dairy environment; dairy product; foodborne pathogen; listeriosis

## 1. Introduction

The dairy industry is among the largest food industries in the world, producing a wide range of perishable and semi-perishable products that are widely demanded. It is estimated that, by 2026, the global dairy trade will exceed USD one billion [1,2]. Ensuring the quality and safety of these products is a major concern for dairy companies. However, due to their highly nutritious nature, dairy products are excellent environments for the growth of a wide variety of microorganisms, and undesirable (spoilage-inducing and pathogenic) microbes can be introduced into the dairy chain during any step, from milking to serving [3–5]. As a result, many milk products have a relatively short shelf life or are associated with cases of foodborne disease, such as listeriosis [1].

*Listeria monocytogenes* is the causative agent of listeriosis, a disease that mainly affects specific groups of individuals. It is especially concerning due to the severity of its sequelae and high mortality rate (up to 30%) [6,7]. Listeriosis is epidemiologically linked to the consumption of foods contaminated with *L. monocytogenes*, especially ready-to-eat (RTE) foods that are stored for extended periods at refrigeration temperatures and do not require heating for consumption [7,8]. The ubiquitous nature of the bacterium, coupled with its ability to thrive in foods and environments where other food-borne pathogens cannot, makes *L. monocytogenes* a major problem throughout the food-production chain, including dairy [9]. Indeed, sporadic cases and outbreaks of listeriosis have been related to the

consumption of dairy products, especially soft, semi-soft, and surface-ripened cheeses; the bacterium is commonly found in dairy products, and its persistence has been demonstrated in dairy-processing environments [10].

The control of *L. monocytogenes* is extremely challenging for the dairy and other food industries, since it persists in harsh environments due to its capacity to adhere and form biofilms, as well as being a pathogen of extreme concern for the health of consumers. Additionally, the detection of *L. monocytogenes* in foods can cause drastic economic losses, with the potential for costly product recalls, laboratory testing, and lawsuits. In this review, we highlight the main characteristics of *L. monocytogenes* and listeriosis, as well as the factors that favor its persistence in processing environments and in dairy products. In addition, we discuss the main dairy products involved in outbreaks of listeriosis since the early 1980s and list the main control measures that can help to prevent the occurrence of this pathogen in foods and food-processing environments.

## 2. An Overview on *Listeria monocytogenes* and Listeriosis

Despite being recognized as a sporadic pathogen since its first isolation from humans in 1929, *L. monocytogenes* only captured the attention of the scientific community in the early 1980s. This was prompted by numerous outbreaks and sporadic cases of foodborne listeriosis that were reported in Canada, the United States, and Europe.

The ubiquitous bacterium *L. monocytogenes* inhabits a wide range of ecological locations, including soil, water, vegetation, drains, feed, sewage, animal, and human feces [11–13]. It is a non-spore-forming facultative, anaerobic, rod-shaped Gram-positive bacterium that is catalase-positive and beta-hemolytic when grown on blood agar. The bacterium is of great concern as it can adapt, survive, and even grow under a wide range of environmental stressors in the food-production industry, such as low temperatures ($-0.4$ °C), as well as being able to withstand repeated freezing and thawing procedures and to survive for long periods under environmental stresses, including a wide pH range (4.4–9.6), water activity below 0.90, high osmotic pressure, ultraviolet (UV) lights, the presence of biocides, and heavy metals [8,11–13]. The exposure of *L. monocytogenes* to one stressor may favor cross-adaptation to subsequent exposure to other stressors [14]. Survival and multiplication in this range of environmental variations allow the pathogen to persist in food-processing environments, survive various food-processing steps, and proliferate in food, making its control extremely challenging.

Furthermore, *L. monocytogenes* is a highly genetically heterogeneous species that exhibits a clonal population structure [15,16]. Based on somatic (O) and flagellar (H) antigens, this pathogen is classified into fourteen serotypes, which are further grouped into four genetic diverse lineages, consisting of specific serotypes, and numerous multilocus sequence types (MLST), which are subcategorized into distinct clonal complexes [17–20]. There are distinct phylogenetic, ecologic, and phenotypic characteristics among the different lineages. These differences are linked to epidemic clones, which are defined as closely related isolates from a probable common ancestor associated with several geographically and temporally unrelated outbreaks of listeriosis [21,22]. Lineage I is the most frequently isolated from clinical human samples and from epidemic outbreaks in most studies [23]. This lineage harbors serotypes 1/2b, 3b, 4b, 4d, 4e, and 7, with serotypes 1/2b and 4b encoding listeriolysin S, a bacteriocin that promotes intestinal colonization by modulating the composition of the host's intestinal microbiota and that is not present in the other lineages [11,23,24]. Lineage II is commonly derived from environmental, agricultural, and food isolates, it is found in both humans and animals, and corresponds to serotypes 1/2a, 1/2c, 3a, and 3c. It often harbors plasmids that contain a plethora of resistance genes that deal with toxic metals, horizontal gene transfer, oxidative stress, and toxic small peptides [11,15,23]. Lineages III and IV are rarer and predominantly identified in animals, with lineage III containing the serotypes 1/2a, 4a, 4b, and 4c, while serotypes 4a and 4c, as well as atypical 4b serotypes, are grouped into lineage IV [11,23]. Recently, *L. monocytogenes* serotype 4h, consisting of hypervirulent isolates from hybrid sub-lineage

II, was reported [18]. Although all the serotypes of *L. monocytogenes* can cause listeriosis, the majority of isolates from human-illness cases belong to serotypes 1/2a, 1/2b, and 4b, with serotype-4b strains being responsible for most foodborne-illness outbreaks and sporadic cases of the disease, suggesting that this serotype has specific virulence properties [9,19].

Compared to other foodborne illnesses, such as salmonellosis, listeriosis is not a common disease in the general population, despite the wide distribution of *L. monocytogenes* in the environment and its relatively high frequency of isolation in a wide variety of foods [7,25,26]. However, it can present a wide range of manifestations in humans, from mild gastroenteritis to neuroinvasive disease, as shown in Table 1. Furthermore, the disease presents a high rate of hospitalizations (up to 97%) and a high fatality rate (up to 30%), even with adequate antimicrobial treatment [6,13,27].

**Table 1.** Disease conditions associated with listeriosis, including the sites of action and major clinical manifestations.

| Diseases | Site of Action | Clinical Manifestations |
|---|---|---|
| Bacteremia | Blood | Fever, chills, myalgia, prodromal symptoms such as diarrhea and nausea |
| Brain abscesses | Central nervous system | Macroscopic brain abscesses, concomitant meningitis |
| Cutaneous listeriosis | Skin infection, conjunctivitis | Low-grade fever, multiple papulopustular skin lesions |
| Gastroenteritis | Gastrointestinal system | Fever, watery diarrhea, nausea, headache, joint, muscle pain |
| Infection in pregnancy | Mother's blood and placenta | In the mother, mild, including fever, back pain, headache, vomiting, diarrhea, muscle aches, sore throat |
| Localized infections | Liver, lungs, joints | Hepatitis, liver abscesses, cholecystitis, peritonitis, splenic abscesses, pleuropulmonary infections, joint infections, co-infectious osteomyelitis |
| Meningitis | Central nervous system: brainstem and meninges | Subacute bacterial meningitis: fever, headache, neck stiffness; Rhombencephalitis: fever, headache, nausea, vomiting |
| Neonatal infection | Neonate | Low birth weight or miscarriage |

Adapted from [28–30].

The average incubation period of listeriosis is estimated to be three weeks, ranging from 1 to 70 days after the consumption of an affected food product [8,11,25]. Since the disease can take a long time to manifest itself after the consumption of contaminated food, it is often very difficult to trace the source of contamination. The minimal dose required to cause clinical infection in humans has not yet been clearly established. In general, foods have low populations of the pathogen, although inadequate processing or storage conditions can lead to the development of large populations [13,31]. The fact that sporadic and epidemic cases of listeriosis are caused by high detected loads of *L. monocytogenes* in foods suggests that it is unlikely that doses of *L. monocytogenes* much lower than $10^3$ CFU/g are responsible for causing the disease, which reinforces the need to minimize human exposure to high populations of the bacterium [20,32,33].

*Listeria monocytogenes* is an intracellular bacterium with the ability to infect a variety of cell types and cross important organic defense barriers, particularly the intestinal, blood–brain and placental barriers [34]. After ingestion, the bacterium reaches the intestine and crosses the epithelial barrier via transcytosis, invading the mesenteric lymph nodes and causing systemic infection [35]. Although invasive listeriosis can occur in healthy

individuals, the main risk factors for developing it include extreme age (newborn or old age), a weakened immune system, and pregnancy. Severe manifestations of listeriosis are due to the presence of the microorganism in the bloodstream (bacteremia), which can progress to sepsis, endocarditis, and central-nervous-tissue infection, through which it can cause meningitis and meningoencephalitis [7,26]. Other clinical manifestations of listeriosis can also occur, especially in patients who have underlying disease comorbidities, and include cutaneous listeriosis, septic arthritis, hepatitis, liver abscess, peritonitis, biliary tract infections, and osteomyelitis [11,28]. In pregnant women, listeriosis may be subclinical, without symptoms, or have "flu-like" symptoms, such as fever, chills, malaise, arthralgia, back pain, and diarrhea [10]. Moreover, *L. monocytogenes* can infect placental tissue, causing chorioamnionitis and bacteremia, and it can also penetrate the placenta and infect the fetus, causing miscarriage, premature delivery, stillbirth, and neonatal septicemia [10,28].

Non-invasive listeriosis, also known as febrile *Listeria* gastroenteritis, is a mild form of the disease that occurs mainly in immunocompetent individuals, rapidly after the consumption of foods with a higher microbial load of *L. monocytogenes* than those reported during outbreaks of invasive listeriosis, with values greater than $10^9$ CFU/g [28,36]. It is characterized by symptoms that include headache, myalgia, fever, nausea, vomiting, diarrhea, abdominal pain, and is usually self-limiting [13,36,37]. However, pre-existing lesions in the gastrointestinal mucosa may allow the translocation of *L. monocytogenes* from the gastrointestinal tract, with the subsequent development of invasive disease [28,38].

### 3. Pathogenesis of *Listeria monocytogenes*

For *L. monocytogenes* to cause an infection, it must first overcome the defense barriers of the human gastrointestinal tract, as it is transmitted orally. The severity of the resulting clinical manifestations depends on the interaction between the pathogen's virulence factors and the host's immune-system responses. This interaction is highly orchestrated, meaning that the pathogen can disseminate through tissues and cause an infection [39,40].

The pathogenesis of *L. monocytogenes* consists of four main steps: (1) the pathogen's survival to the attacks of the first nonspecific host-defense barriers; (2) its adhesion to and invasion of gut cells; (3) the lysis of the vacuole; and (4) the pathogen's intracellular multiplication and intercellular spread to adjacent host cells (translocation). A detailed description of each step is provided by Quereda et al. [38]. More specifically, *L. monocytogenes* encounters an acidic environment after it is ingested through contaminated food. The pathogen possesses intracellular pH-regulation systems that enable it to adapt and survive in low-acidity conditions (for a more detailed explanation, please refer to Smith et al. [41]). Individuals treated with gastric-acid suppression with H2-antagonist and proton-pump inhibitors have an increased risk of developing listeriosis [38,42]. In addition to stomach acidity, the pathogen also faces challenges posed by bile salts, nonspecific inflammatory defenses, and host proteolytic enzymes [11]. The gut microbiota is also an important barrier against the pathogen, not only because of microbial competition (and resistance to colonization) but also because it can activate immune responses that limit the spread of listerial infection [38,43,44]. In fact, Becattini et al. [45] presented a study that demonstrated how the use of antimicrobial therapy increased susceptibility to *L. monocytogenes* infection. This emphasized the significance of diverse intestinal microbiota in defending against this pathogen. Nevertheless, certain hypervirulent *L. monocytogenes* strains produce LLS, a bacteriocin encoded by the *lls* gene in the *Listeria* Pathogenicity Island 3 (LIPI-3). During intestinal colonization, LLS is overexpressed and targets specific Gram-positive bacteria, including species of the *Alloprevotella*, *Allobaculum*, and *Streptococcus* genera. This results in the alteration of the intestinal microbiota composition, facilitating the efficient colonization and invasion of deeper tissues by the pathogen [24,38,44].

*Listeria monocytogenes* not only survives the harsh environment of the gastrointestinal tract but also invades epithelial cells and spreads to other organs, including the liver, spleen, blood, central nervous system (CNS), placenta, and fetus [46]. Upon arriving in the intestine, the pathogen adheres to the intestinal mucosa and initiates the cell-adhesion stage

in both phagocytic and non-phagocytic cells [47]. Internalin A (InlA), an 800-amino-acid protein containing 15 leucine-rich repeats (LRR), is a primary virulence factor associated with the invasion of non-phagocytic intestinal cells. Its cellular receptor is E-cadherin, a transmembrane protein involved in cell adhesion. The interaction between InlA and E-cadherin allows the pathogen to enter the host cell and evade immune responses. The bacterium also secretes other LRR proteins, including internalin B (IntB), which are involved in the internalization of the pathogen into intestinal cells (for more details, please refer to Pizarro-Cerdá and Cossart [40]). In professional phagocytic cells, such as intestinal M cells, the internalization process differs from those of non-phagocytic cells and is independent of InlA, occurring instead through the process of phagocytosis [48].

As mentioned above, *L. monocytogenes* utilizes its cellular ligands (IntA, IntB) to adhere to intestinal epithelial tissue and internalize into the host cell. Once inside, the bacterium secretes listeriolysin O (LLO), a toxin that causes the lysis of the internalizing vacuole. Listeriolysin O is accompanied by two phospholipases C (PLCs), PlcA and PlcB, which work with LLO to form pores that disrupt the vacuolar membrane, allowing *L. monocytogenes* to escape into the host cell. In conjunction with the activities of LLO and PLC, quorum sensing has also been associated with vacuole escape [46,48–50]. Quorum sensing is a form of bacterial communication that occurs through signaling molecules called autoinducers, depending on the cell density of the environment [44,48]. The detection of and response to these signals trigger bacterial gene expression in a regulated and joint manner, helping in their survival. The accumulation of these molecules inside the membrane compartment mimics the proliferation of the microorganism and leads to the rupture of the vacuole [48].

Free in the host's cytoplasm, *L. monocytogenes* multiplies and initiates the process of translocation to neighboring cells. To this end, the bacterium secretes an actin nucleating factor (ActA), which is involved in the polymerization of actin filaments, forming a thruster tail in the pathogen, which drives bacterial propulsion, allowing the spread of the pathogen from cell to cell [46,49]. Interestingly, Cheng et al. [51], demonstrated that, in the absence of PLC, actin polymerization is necessary for the multiplication of *L. monocytogenes* in macrophages. Bacterial motility is also an important mechanism through which the pathogen evades the immune system as it interferes with the formation of autophagosomes, forming a physical barrier composed of molecules from the host itself, preventing the pathogen's recognition by the innate immune system, which is necessary for autophagy [51]. However, the ActA-mediated mechanism that prevents autophagy remains unclear [51]. After translocation, *actA* expression gradually decreases, which causes the cessation of actin polymerization on the bacterial surface. Thus, the bacteria that remain inside the cells, without cover from the actin filaments, are internalized in lysosomal vacuoles, where they remain in a viable but non-culturable (VBNC) state, increasing the incubation period in the asymptomatic transmitting host [52].

In the secondary infected cell, a new double-membraned vacuole originating in the donor and recipient cells surrounds the microorganism, at which point the process restarts, with the bacteria spreading until reaching the infection target organs [40,49,53]. Bacteria that cross the intestinal barrier are carried through the lymph nodes and blood to the mesenteric lymph nodes, spleen, and liver. Although resident liver macrophages play an important role in containing the infection, hepatocytes are the preferred sites for *L. monocytogenes* multiplication [44,54]. During the initial stages of infection, hepatocytes respond to the presence of this bacterium by releasing chemokines, consequently mobilizing neutrophils, and by inducing apoptosis, which results in the formation of microabscesses [55]. If the infection is not controlled in the liver, the intense bacterial proliferation can result in the release of bacteria into the bloodstream, leading to bacteremia [13]. Although there are many studies on the invasion process of *L. monocytogenes*, the mechanisms underlying non-invasive listeriosis remain unclear. In healthy individuals, *L. monocytogenes* infections are usually limited to the extracellular compartment within the intestinal lumen, with self-limiting intestinal manifestations [38,46].

Antibiotic therapy is recommended for severe cases of listeriosis. As *L. monocytogenes* is an intracellular pathogen, the antibiotic of choice to treat listeriosis must penetrate host cells, diffuse, and remain stable in the intracellular environment, which can reduce the effectiveness of antibiotics to as little as 30% [44,55]. The antibiotics most commonly used for treating listeriosis include ampicillin, penicillin, and amoxicillin, but other antibiotics can be also used [28–30]. The duration of treatment is variable and depends on the severity of the disease and the patient's clinical history, as in the case of immunocompromised patients, who usually need longer treatment [29].

The resistance of *L. monocytogenes* to penicillin, ampicillin/amoxicillin, gentamicin, and trimethoprim-sulfamethoxazole has increased recently, which is concerning [43,56–58]. This increase in *L. monocytogenes* that are resistant to antibiotics is probably linked to the excessive prescription of medicines to treat human infections, the widespread use of antimicrobials as growth promoters in animal feed, and the increases in global trade and travel, which support the dissemination of resistant microorganisms among countries [11]. In a further complication, biocide-tolerant strains of *L. monocytogenes* have been found to be resistant to antibiotics that are commonly used for the treatment of listeriosis [52]. Biocides are used in the hygiene protocols in food-processing industries to control and prevent contamination by pathogens such as *L. monocytogenes*. However, sublethal concentrations of biocides may remain after the sanitization process, exerting selective pressure on *L. monocytogenes*, which can lead to tolerant strains or the selection of resistant strains [52,59]. Environmental-stress tolerance occurs due to phenotypic changes, such as the VBNC state, cell-surface modifications, biofilm formation, and the activation of the efflux-pump system, while genotypic changes are related to the acquisition of mobile genetic elements (MGEs), including prophages, plasmids, and transposons carrying resistance genes [52].

## 4. *Listeria monocytogenes* in the Dairy Environment

The wide distribution of *L. monocytogenes* stems from its unique ability to survive and persist across a broad range of unfavorable environmental conditions, which gives it advantages over the competitive microbiota and most other non-spore-forming foodborne pathogens that affect humans. Thus, it is not surprising that *L. monocytogenes* is often found in dairy environments, both on farms and in producing plants (Table 2). Once it establishes itself in the environment, the control of this pathogen is extremely difficult and, if proper control measures are not taken, the bacterium can persist in the production facility for years, creating a potential route for cross-contamination into the food chain.

Although the diversity of this species has not been completely explored, the development of genotypic typing systems with high discriminatory power has allowed the subtyping of *L. monocytogenes* at various levels of phylogenetic classification, from genetic lineages to epidemic clones. This has facilitated the development of epidemiological investigations to better understand the pathogen's ecology and transmission, thus enabling the identification of possible sources of contamination and persistence [21,22,60]. Using these tools, it is also possible to systematically compare isolates of *L. monocytogenes* from different sources with those associated with human cases of listeriosis to confirm the existence of a clonal relationship. Some of the molecular methodologies successfully used to elucidate *L. monocytogenes* genetic resemblance include macrorestriction with specific restriction enzymes combined with pulsed-field gel electrophoresis (PFGE), ribotyping, MLST, whole-genome sequencing (WGS), and core-genome multilocus sequence typing (cgMLST) [19,61,62].

**Table 2.** Detection of *Listeria monocytogenes* in dairy environments.

| Dairy Environments | Country | Site of Isolation | Serotypes | References |
|---|---|---|---|---|
| Dairy plants | Austria | Drain water, drain biofilm, floor drains | NS | [63] |
| | Sweden | Milk from farm bulk tanks, raw milk in storage | 1/2a, 1/2b | [64] |
| | Ireland | Raw milk, food, floors, steps, drains | NS | [65] |
| | Portugal | Processing-plant area | 1/2a, 1/2b, 1/2c, 4b | [66] |
| | Austria | Drains, shoes, floors | 1/2a, 1/2b, 3a, 3b, 4b, 4d, 4e, 7 | [67] |
| | | Brine, food, and non-food-contact surfaces, | 1/2b, 1/2c, 4b | [68] |
| | | Farm bulk tank | 1/2a | [69] |
| | Brazil | Processing-room floor, raw milk, cooling tank, processing room drain | 4b, 4d, 4e | [70] |
| | | Processing-room drain, brine-room floor | 1/2b, 3b, 7 | [70] |
| | | Cooling-chamber drain, floors and platforms of processing rooms, plastic crates, gloves, brine | 1/2b, 1/2c, 4b | [71] |
| | | Floor drain | NS | [59] |
| | Spain | Conveyor belt, floor, food soil, packaging bench, conveyor belt | 1/2a-3a, 1/2b, 1/2c-3c, 3b-7, 4b-4d-4e | [62] |
| | United States | Bulk-tank milk, milk filters | 1/2a, 1/2b, 4a, 4b | [72] |
| Dairy farms | Canada | Farm animals, water supply, plant surfaces, drains, air vents | 4b/4b | [73] |
| | United States | Fecal-grab samples | 1/2a, 1/2b, 4b | [22] |
| | Iran | Bulk-tank milk | 1/2a, 1/2c, 3a, 3c, 4b, 4d, 4e | [74] |
| | Slovenia | Raw milk, bulk tank or pooled milk, silage, feces, water | NS | [16] |
| | Finland | Bulk-tank milk, filter sock, barn environment | 1/2a, 1/2b, 1/2c, 3a, 3b, 4b, 4d, 4e, 7 | [75] |
| | Brazil | Bulk-tank milk, milk-filter socks, milk-room floors | 2a, 4b | [76] |
| | Norway | Bulk-tank milk, milk filters, feces, feed, teats, teat milk | NS | [77] |
| | Spain | Forage, water, raw tank milk, milk filters, fresh feces, stored manure, soil | 2a, 2b, 4b | [19] |

NS: not specified.

Several studies have confirmed that dairy animals and rural environments harbor a wide diversity of *L. monocytogenes*, including the genotypes that cause human listeriosis, thus contributing to the dissemination of the pathogen throughout the food chain [16,19,21,22,78]. However, the ecology of *L. monocytogenes* within farm environments is now well understood. Poorly fermented silage seems to be the main reservoir that introduces *L. monocytogenes* into dairy-farm environments, although other on-farm sources of infection (e.g., water, feeders, bedding) are also possible [19,79]. In ruminants (cattle, goats, sheep, buffalo), the typical manifestations of listeriosis are similar to those that occur in humans, and animals can present encephalitis, septicemia, and uterine infections that often result in abortion, which can lead to significant economic losses [80]. Both clinically infected animals and asymptomatic carrier animals can shed *L. monocytogenes* into farm environments through their feces, through which the microorganism can contaminate the udders and cause mastitis, as well as spreading to milking utensils, filters, and bulk storage containers, leading to the contamination of raw milk [19,77,81]. The same *L. monocytogenes* genotype can occur on a farm over long periods, whereas other subtypes are identified only sporadically. It is not yet clearly established whether this is a result of the repeated reintroduction of a specific *L. monocytogenes* subtype from external contaminating sources or from its continuous presence (true persistence) within farm environments, or both [60,75].

The persistence of *L. monocytogenes* in dairy environments is associated with the different genotypic and phenotypic characteristics of the strains that facilitate their survival and growth [62,82]. Mobile genetic elements, such as prophages, plasmids, and transposons, are common among *L. monocytogenes* isolates from food-processing facilities, and these

elements may contain genes involved in tolerance to environmental stresses, such as heat and osmotic shock and acid stress, as well as resistance to chemical compounds and biocides [23,82,83]. These genes can be horizontally transferred and are associated with the long-term survival of *L. monocytogenes* in food-processing lines [82]. Castro et al. [83] sequenced the genomes of 250 *L. monocytogenes* isolated over a three-year period from three Finnish dairy farms. The authors found that MGEs were more abundant among persistent than among non-persistent *L. monocytogenes* isolates, conferring an ecological advantage for persistence in the environment and constituting a reservoir of diverse MGEs that could spread downstream in the food chain.

Another important characteristic that is linked to the persistent presence of *L. monocytogenes* in food-processing lines is its ability to form biofilms under unfavorable environmental conditions. Biofilms are highly organized communities created by microbial cells that attach to various surfaces and are held together by a self-generated, thick web of extracellular polymeric substances. This layer protects the cells from challenging conditions, such as limited access to nutrients or exposure to antimicrobial agents [1,84,85]. After contaminating milk-handling equipment, milk lines, or milk-storage tanks, *L. monocytogenes* can adhere to these equipment surfaces, forming biofilms that make it more difficult to eliminate it from the dairy environment. The adhesion step is facilitated by the presence of milk and other organic residues on the surfaces of utensils and equipment [86]. For instance, Lundén et al. [87] found that persistent *L. monocytogenes* strains have greater adhesion to food-contact surfaces under short contact times than non-persistent strains, which possibly influences the onset of persistent plant contamination. Borucki et al. [88] also found that persistent strains of *L. monocytogenes* isolated from bulk milk tanks are better biofilm formers than non-persistent strains. On the other hand, findings by Lee et al. [14] suggest that high adhesion to surfaces may not be the main prerequisite for *L. monocytogenes*' persistence in food-producing environments, which highlights the importance of other traits associated with food-chain stressors, such as increased resistance to disinfectants or desiccation treatments, for the persistence of the pathogen.

It is also important to highlight that, in nature, biofilms are rarely formed by a single bacterial species, so *L. monocytogenes* can present different types of interaction, both cooperative and competitive, with the microbiota residing in colonized microenvironments [89]. The type of microbial interaction and the specific particularities of the pathogen strain have a direct influence on *L. monocytogenes*' proliferation and survival in food-production environments [90]. However, once it is sheltered in mono or multi-species biofilms, *L. monocytogenes* becomes more difficult to eradicate due to its strong resistance to environmental threats.

Latorre et al. [87] and Latorre et al. [86,91] documented that biofilms identified on milking equipment contributed to the persistence of the same genotype of *L. monocytogenes* for years in a bulk milk tank and in in-line milk-filter samples from a farm in the United States. Castro et al. [75] investigated the molecular epidemiology of *L. monocytogenes* in samples of bulk tank milk, milk filter sock, and barn environments from three Finnish dairy cattle farms during 2013 to 2016. These authors observed that persistent *L. monocytogenes* genotypes isolated from the milking system or cows' udders were more likely to contaminate bulk-tank milk and milk-filter stocks than genotypes that occurred only in other parts of the farm. Furthermore, they reported that the environmental sites with the lowest hygiene scores had the highest prevalence of *L. monocytogenes*, which was in accordance with the findings of Fox et al. [92], who evaluated dairy farms in Ireland and identified a correlation between the hygiene standards on the farm and the occurrence of *L. monocytogenes*. Using culture-independent methods, Weber et al. [84], in Germany, identified sequences of clones assigned to *L. monocytogenes* in biofilms sampled from milking machines about four hours after the cleaning-and-sanitization process, which was indicative of the permanent presence of this pathogen in the farm's milking system and indicated that the cleaning process alone may not be sufficient to remove biofilms from milking machines.

Regarding the contamination of dairy manufacturing facilities, *L. monocytogenes* can be introduced into these environments through a variety of routes, including incoming

raw milk, which is the main source, as well as employees (e.g., workers' boots), crates or transport vehicles, airflow, and traffic flow, among others [93,94]. After entering the production line, the pathogen can rapidly colonize sites that are difficult to access for cleaning and sanitization procedures, such as cracks and crevices present in floors, walls, drains, pipes, conveyor belts, mixers, dicing machines, slicers, freezers, condensers, gaskets, carts, packaging machines, and so on [60,93,94]. Once housed in these niches, *L. monocytogenes* can form biofilms and persist in food-production lines for months or years, which facilitates its spread to downstream points within the processing lines, leading to the contamination of dairy products at different stages of processing.

Melero et al. [62], in Spain, investigated the presence and persistence of *L. monocytogenes* in a newly established dairy-processing facility. Over the course of a year, the authors made regular visits to two buildings of the same dairy facility, 13 km apart. The initial stages of cheese production (receiving milk, processing, curdling, salting, and stacking for transport) were carried out in the old building, while the final processing stages (maturation, slicing, modified atmosphere packaging, and cheese grating) were carried out in the newly opened building. *Listeria monocytogenes* was found on all but one visit to the old building. However, the first isolation of *L. monocytogenes* in the new building took place only on the third visit, nine months after the start of production activity, and occurred in the cheese-grating area, from where it spread to other areas of the facility and corresponded to the main type of PFGE that persisted throughout the study period. The identification of persistent strains of *L. monocytogenes* at different sites in dairy-processing plants over time, even after cleaning and disinfection procedures, not only draws attention to the ongoing risk of product contamination during and after processing, which highlights the need for environmental testing as a control measure for *L. monocytogenes* in industrial environments, but also emphasizes the need to monitor raw materials, to prevent the bacterium from entering processing plants [60,66,91].

Kells and Gilmour [65] studied two milk-processing facilities for over one year in Northern Ireland and repeatedly detected *L. monocytogenes* in different samples, including raw milk, food, and environmental areas, such as floors, steps, and drains. However, the authors did not use genetic-subtyping methods to verify whether the *L. monocytogenes* strains were persistent in the studied dairy plants. The use of molecular methods to enable environmental monitoring is key to identifying persistent sources of contamination and routes of transmission of *L. monocytogenes* through the food chain [92,95]. A study conducted over a three-year period [96], using ribotyping, found clear evidence of the persistence of distinct ribotypes of *L. monocytogenes* at different sites (drains, floor areas, equipment surfaces, walls, and doorways) on a sheep farm and its associated farmstead dairy-production facility, in the United States. Among the eight different ribotypes of the pathogen found, only one was present both on the farm and in the processing unit, and the data obtained in the study indicated the limited transfer of *L. monocytogenes* between the evaluated environments.

The presence of environmentally persistent *L. monocytogenes* in final products has also been evidenced in several studies. Over the course of six months, Kabuki et al. [97], in the United States, carried out a study with environmental and Latin-style fresh-cheese samples from three processing plants. The same *L. monocytogenes* ribotype was found in two of the processing plants, and in one of them, the ribotype was persistent and widespread; it was also found in the final product, which was indicative of post-processing contamination. Curiously, the same *L. monocytogenes* ribotype found by these authors had already been associated with a multistate outbreak of listeriosis in the United States, which involved the consumption of hot dogs and deli meats [98]. Barancelli et al. [68] collected cheese and environment samples from three small-scale dairy plants in Brazil for about a year. Using serotyping and PFGE, the authors repeatedly found indistinguishable profiles of *L. monocytogenes* in two of them, indicating the persistence of the bacteria in both dairy-processing plants. Moreover, in one of the dairy plants, the bacterium was found in the finished product. Using multi-virulence-locus sequence typing (MVLST), Fillipelo

et al. [99], in Italy, identified a clone of *L. monocytogenes* (virulence type 14—VT14) isolated from Gorgonzola cheese and environmental samples collected in a single processing plant from 2004 to 2007, indicating the persistence of this clone in the production facility. It is noteworthy that the VT14 had previously been isolated from sporadic cases of listeriosis in Italy and was also identified in an outbreak of listeriosis linked to the consumption of chocolate milk [99,100]. The literature is rich in reports on the persistence of the same genetic profile of *L. monocytogenes* for prolonged periods in different locations within dairy- and other food-processing lines, leading to the contamination of final products, which poses a risk to the health of consumers [66,95,101,102]. In fact, there were even confirmations of outbreaks of listeriosis linked to the post-processing contamination of dairy products, such as the outbreak that occurred in Canada in 2015–2016 (see the Outbreaks section, below), in which the presence of the *L. monocytogenes* strain responsible for the outbreak was confirmed by PFGE in a chocolate-milk line, inside a post-pasteurization pump used for the product, and on non-food-contact surfaces [103].

## 5. *Listeria monocytogenes* in Dairy Products

According to Lee et al. [104] the occurrence of *L. monocytogenes* in raw milk is highly variable, ranging from 0 to 26% in America, from 0 to 28.7% in Europe, from 0 to 22.4% in Africa, and from 0 to 50% in Asia and Oceania. Thus, raw milk is clearly a risk factor for listeriosis, and it is strongly recommended that milk and its derivatives are not consumed raw, especially by people belonging to risk groups [105] However, the production of dairy products using raw milk is still a common practice that can result in new cases and outbreaks of listeriosis (see next section).

Pasteurization is a widely accepted method for ensuring the safety of milk and its derivatives. When carried out correctly, this process is effective in eliminating *L. monocytogenes* from dairy products [106]. However, the risk of L. monocytogenes contamination can persist even after pasteurization due to inappropriate temperatures or equipment failure during the process, or contamination during subsequent production steps. As a result, pasteurized dairy products may still contain this microorganism [87,104,106]. Studies indicate that L. monocytogenes within leukocytes in milk can survive if the pasteurization temperature is not adequate [107].

The occurrence of *L. monocytogenes* is also highly variable in different processed dairy products, particularly in soft cheeses (Table 3). Soft cheeses constitute a heterogeneous group of products that include fresh cheeses, such as Hispanic-style or Latin-style cheeses (queso fresco, cotija, Minas Frescal), and soft-ripened cheeses, such as brie, Camembert, and other mold- and smear-ripened cheeses [108]. Fresh cheeses do not pass through a maturation process, while soft-ripened cheeses can be ripened for up to two months. The group of soft cheeses has some common characteristics, such as high moisture content ($\geq$50%), low acidity, low salt content, and the need for refrigeration for safety [109,110]. Thus, the intrinsic characteristics of soft and soft-ripened cheeses allow the growth of *L. monocytogenes* even when stored (or ripened) under suitable refrigeration conditions [27,109]. Furthermore, soft cheeses are RTE products that are usually consumed without additional heat treatment, which also contributes to the involvement of these foods in listeriosis cases and outbreaks.

The maturation of hard and semi-hard cheeses, such as cheddar and parmesan, modifies their initial intrinsic characteristics, and the reduction in moisture content and water activity, associated with the activity of starter and non-starter cultures, creates a hostile environment that limits the development of several pathogens, including *L. monocytogenes* [109,111]. Thus, since the aging process mitigates the risk of the transmission of listeriosis, even when raw milk is used for cheese manufacturing, these products are less frequently contaminated with *L. monocytogenes* than their soft, unripened counterparts [109,112]. On the other hand, although ripened cheeses generally do not allow the growth of *L. monocytogenes*, and even decreases the initial populations of the bacteria are found, depending on the aging time of the hard cheese, the bacteria can still survive in

these products [113]. Bellio et al. [114] demonstrated that the pathogen survived for a prolonged period in artificially contaminated 80-day-ripened cheese stored at 4 °C.

In addition to milk and cheese, other dairy products in which *Listeria* occurs include ice cream and butter [9,115,116]. Chen et al. [117] reported a massive contamination with *L. monocytogenes* of ice cream from a production line in the United States. This ice cream was the vehicle of an outbreak of listeriosis (see next section), and 99.4% of the 2.320 samples of ice cream produced on seven different dates that were evaluated to confirm the connection between the food and the outbreak were contaminated with *L. monocytogenes*. On the other hand, although other *Listeria* species were present in ice cream (14 out of 30 samples), Ewida et al. [118], in Egypt, did not find *L. monocytogenes* in the product (0 in 30 samples). However, despite the great variation in relation to the occurrence of *L. monocytogenes* in ice cream, it is undeniable that its presence in the product may represent a risk for consumers, since it was demonstrated in a risk-assessment study that the pathogen can survive for 36 months in ice cream stored at −20 °C, without a significant decrease in the initial population [119]. Furthermore, this ice-cream outbreak suggests that human listeriosis cases may occur after the widespread distribution of products that are unable to support the growth of this pathogen but are persistently contaminated at low levels, if consumed by highly susceptible populations [33].

The cross-contamination of dairy products with *L. monocytogenes* is an ongoing risk throughout the food supply chain, and it can also occur at retail establishments. Deli countertops and slicers, for example, are recognized as major sources of the contamination of dairy and other food products with *L. monocytogenes* [98,120]. Russini et al. [121] described a small outbreak (four cases) of listeriosis in a hospital in Italy, where the meat slicer was the source of food contamination with the pathogen. Currently, an investigation into a listeriosis outbreak is underway in the United States, with slicers believed to be the sources of *L. monocytogenes* contamination in various food products [122].

**Table 3.** Occurrence of *Listeria monocytogenes* in different types of dairy product.

| Dairy Product | Country (Regional Name of the Product) | Number of Samples | Occurrence (%) | References |
|---|---|---|---|---|
| Milk and milk products | European Union | 2479 | NA | [123] |
| Butter | Belgium | 603 | 66.0 | [116] |
| | United Kingdom | 33 | 0.4 | [124] |
| Cream cheese | Italy | 108 | 1.9 | [125] |
| Fresh cheese | Austria * | 25 | 4.0 | [126] |
| | | 27 | 0.0 | [126] |
| | Italy * | 31 | 12.9 | [127] |
| | | 15 | 6.7 | [91] |
| | | 258 | 3.5 | [128] |
| | | 60 | 0.0 | [129] |
| | | 149 | 3.4 | [130] |
| | Mexico * | 100 | 6.0 | [131] |
| | | 16 | 6.3 | [132] |
| | | 16 | 37.5 | [132] |
| | | 75 | 9.3 | [133] |
| | Spain * | 78 | 1.3 | [134] |
| | Sweden * | 78 | 0.0 | [135] |
| | United States * | 204 | 2.0 | [136] |
| Ice cream | Egypt | 40 | 7.5 | [115] |
| | | 30 | 0.0 | [118] |
| | United States | 2320 | 99.0 | [117] |
| Semi-hard cheese | Brazil (Canastra) | 78 | 1.0 | [112] |
| | Brazil (Serro) | 256 | 0.0 | [137] |
| | Turkey (Tulum) | 250 | 4.8 | [138] |

**Table 3.** *Cont.*

| Dairy Product | Country (Regional Name of the Product) | Number of Samples | Occurrence (%) | References |
|---|---|---|---|---|
| Semi-soft cheese | Italy (Blue-veined) | 120 | 55.0 | [139] |
| | Italy (Gorgonzola) | 1489 | 2.1 | [140] |
| | Italy (Mozzarella) | 186 | 0.0 | [125] |
| | Turkey (Homemade) | 142 | 9.2 | [141] |
| Soft cheese | Austria * | 233 | 4.7 | [126] |
| | Belgium * | 32 | 3.1 | [142] |
| | Bulgaria * | 63 | 0.0 | [143] |
| | Czech Republic * | 387 | 5.2 | [144] |
| | Egypt (Cottage) | 50 | 0.0 | [145] |
| | Ethiopia (Cottage) | 100 | 1.0 | [146] |
| | European Union * | 3452 | 0.5 | [147] |
| | Greece * | 10 | 40.0 | [148] |
| | Greece (Panel) | 137 | 0.0 | [127] |
| | Iraq * | 50 | 2.0 | [149] |
| | Italy (Panel) | 444 | 4.7 | [131] |
| | | 894 | 2.1 | [135] |
| | Portugal * | 49 | 14.3 | [129] |
| | Spain (Panel) | 163 | 0.0 | [134] |
| | Sweden * | 525 | 0.4 | [150] |
| Soft fresh cheese | Brazil (Minas Frescal) | 55 | 11.0 | [69] |
| | Egypt (Kareesh) | 30 | 0.0 | [151] |
| | Italy (Burrata) | 404 | 0.0 | [152] |
| | Italy (Ricotta) | 30 | 0.0 | [91] |
| | Mexico (Adobera) | 100 | 12.0 | [131] |
| | | 16 | 18.8 | [132] |
| | Morocco (Jben) | 96 | 4.2 | [153] |
| Soft-ripened cheese | Italy (Brie) | 300 | 1.0 | [154] |
| | Italy (Camembert) | 178 | 0.0 | [154] |
| Traditional whey | Morocco | 52 | 5.7 | [155] |
| White brined cheese | Jordan | 350 | 12.0 | [156] |

* Types of cheeses not clearly identified in the studies.

## 6. Outbreaks of Listeriosis Linked to Dairy Products

Prior to the listeriosis outbreaks that occurred in the early 1980s, it was not clear whether the disease was transmitted through food [157]. However, the German researcher Heinz P. R. Seeliger already suspected the involvement of unpasteurized milk and other dairy products (sour milk, whipped cream, and cottage cheese) during the first reported outbreak of listeriosis in the world, which occurred in Germany between 1949–1957 [87] (Table 4). The cause of this outbreak has not been clarified because, at the time, it was not possible to correlate the infection with the ingestion of food contaminated with *L. monocytogenes*, and a large number of the cases in humans were mainly associated with pregnancy, with no relevant source of infection found [87,158,159]. Dairy products (pasteurized milk) are also under consideration as the possible sources of an outbreak caused by *L. monocytogenes* serotype 4b that afflicted 23 patients in a hospital area in Boston in 1979 [160]. In addition to these two possible outbreaks of listeriosis associated with the consumption of dairy products, Robertson [156] cites the work of Potel [161] as the first study to have observed a direct correlation between a case of listeriosis and the ingestion of raw milk, due to the finding of the same *Listeria* serotype both in a milking cow and in the pregnant woman who miscarried after the ingestion of the animal's milk.

Due to the efforts of countless researchers from around the world, it is now well established that both raw and pasteurized milk, as well as their derived products, provide excellent growth conditions for *L. monocytogenes*, and these products are commonly

involved in cases and outbreaks of listeriosis, as can be observed in Table 4. Although the exact number of listeriosis cases related to outbreaks since the 1980s is not known, it is believed to be higher than the reported figure [87,162]. The first well documented outbreak of listeriosis linked to dairy products occurred in the United States in 1983, and pasteurized milk was the incriminated food [157]. In this outbreak, there were forty-nine cases of the disease, seven in fetuses or infants, and forty-two in immunocompromised patients, with a death rate of 29% (14 patients). Shortly thereafter, an outbreak attributed to the consumption of Mexican-style fresh cheese occurred in 1985, also in the United States, causing 142 cases of the disease with 48 deaths (a death rate of 33.8%) [163]. These outbreaks were crucial in clarifying the role of food in the spread of listeriosis.

Currently, with the development and association between different molecular methodologies, including approaches based on next-generation sequencing (NGS), greater agility in tracking and elucidating listeriosis outbreaks is possible. The molecular typing of *L. monocytogenes* isolates from food and clinical cases makes it possible to link food and disease cases, helping to elucidate original source of contaminations. For many years, PFGE has been considered the gold standard for the laboratory analysis of food and clinical isolates in listeriosis-outbreak investigations, since it is very useful in detecting listeriosis clusters [68]. However, since PFGE does not allow the measurement of phylogenetic relatedness, closely related *L. monocytogenes* isolates may have different PFGE profiles, while unrelated isolates may be indistinguishable when using this methodology [100,164]. Currently, the WGS has been shown to be more relevant from a phylogenetic point of view, and has enhanced listeriosis-outbreak surveillance [165].

**Table 4.** Outbreaks of listeriosis linked to the consumption of milk and dairy products.

| Period | Dairy Product | Country | Cases | Deaths (Stillbirths) | References |
|---|---|---|---|---|---|
| 1949–1957 | ♦ Raw milk | Germany | About 100 | NA (NA) | [87] *apud* [166] |
| 1979 | ♦ Pasteurized milk | United States | 23 | NA (NA) | [160] |
| 1983 | Pasteurized milk | United States | 49 | 12 (2) | [157] |
| 1983–1987 | Pasteurized soft cheese | Switzerland | 122 | 33 (NA) | [167] |
| 1985 | * Soft cheese | United States | 142 | 48 (20) | [163] |
| 1986 | Pasteurized milk | Austria | 28 | 5 (0) | [25] |
| 1989–1990 | * Hard cheese | Denmark | 26 | 6 (NA) | [42] |
| 1994 | Pasteurized chocolate milk | United States | 45 | 0 (0) | [168] |
| 1995 | Raw soft cheese | France | 20 | 0 (4) | [169] |
| 1998–1999 | Pasteurized butter | Finland | 25 | 6 (6) | [170] |
| 2000 | Raw soft cheese | United States | 13 | 0 (5) | [171] |
| 2001 | * Soft cheese | Sweden | 27 | 0 (0) | [172] |
| 2001 | ** Cheese | Japan | 86 | NA (NA) | [173] |
| 2002 | Pasteurized soft cheese | Canada | 135 | NA (NA) | [73] |
| 2005 | Pasteurized soft cheese | Switzerland | 10 | 3 (2) | [174] |
| 2006–2007 | Pasteurized acid-curd cheese | Germany | 189 | 26 (NA) | [175] |
| 2006–2014 | * Soft cheese | Italy | 306 | NA (NA) | [176] |
| 2007 | Pasteurized milk | United States | 5 | 3 (1) | [177] |
| 2008 | Pasteurized milk cheese | Canada | 38 | 2 (3) | [178] |
| 2008–2009 | Pasteurized milk cheese | United States | 8 | 0 (2) | [179] |
| 2009 | * Acid-curd cheese | Germany and Austria | 14 | 4 (0) | [180] |
| 2009–2010 | * Acid-curd cheese | Austria | 34 | 8 (NA) | [181] |
| 2009–2012 | Pasteurized soft cheese | Portugal | 30 | 11 (1) | [182] |
| 2011 | Pasteurized hard cheese | Belgium | 12 | 2 (0) | [183] |
| 2012 | Pasteurized soft cheese | Spain | 2 | 0 (0) | [184] |
| 2012 | * Soft cheese | United States | 22 | 4 (1) | [185] |
| 2013 | Raw soft cheese | United States | 5 | 1 (1) | [186] |

**Table 4.** *Cont.*

| Period | Dairy Product | Country | Cases | Deaths (Stillbirths) | References |
|---|---|---|---|---|---|
| 2014 | Soft cheese | United States | 8 | 1 (NA) | [187] |
| 2014 | Raw milk | United States | 2 | 1 (0) | [188] |
| 2014 | Pasteurized ice cream | United States | 2 | 0 (0) | [189] |
| 2014 | Unpasteurized chocolate milk | United States | 2 | 1 (0) | [190] |
| 2014–2015 | * Ice cream | United States | 4 | NA (0) | [33] |
| 2015–2016 | Pasteurized chocolate milk | Canada | 34 | 4 (NA) | [103] |
| 2021 | Pasteurized soft cheese | United States | 13 | 1 (2) | [191] |
| 2021–2022 | * Ice cream | United States | 28 | 1 (NA) | [192] |
| 2022–ongoing | *** Sliced cheese | United States | 16 | 1 (NA) | [162] |
| 2023–ongoing | *** Semi-soft cheese | United Kingdom | 3 | 1 (NA) | [182] |

♦ Probable outbreaks caused by eating dairy products contaminated with *L. monocytogenes*; * no information on pasteurization; ** no information on the specific name of the cheese; *** ongoing outbreaks under investigation; NA: data not available.

An illustrative case of the discriminatory power of WGS can be seen in the study carried out by Chen et al. [164]. These authors used pooled epidemiological evidence, PFGE data, and multiple WGS analyzes to track a listeriosis outbreak in the United States in 2013, involving fresh cheese [184]. This outbreak was reported to affect one patient from California and seven from Maryland. The *L. monocytogenes* isolated from the patients were from serotype 1/2b, with indistinguishable PFGE profiles. The data from the epidemiological investigation indicated that the patients interviewed in Maryland had consumed Hispanic-style cheese that was produced by company A. However, it was not possible to collect the food history of the California patient. In order to complement the epidemiological data, WGS was used to determine the genetic correlations between the five available *L. monocytogenes* isolates from the outbreak. In addition to these isolates, the following were included: one isolate from a cheese sample collected in New York in 2012 that, according to the PulseNet database, showed an indistinguishable PFGE pattern from those of the outbreak isolates; forty-eight isolates from food and environmental samples from company A, collected in 2014; and one environmental isolate from company B, collected during regular surveillance, three months after the outbreak investigation. Upon the completion of the WGS analysis and routine surveillance, the analysis allowed the exclusion of the *L. monocytogenes* isolate from the California patient from this listeriosis outbreak and also distinguished the PFGE-matched isolate collected from the New York food, even though both showed PFGE profiles that were indistinguishable from those of the confirmed outbreak isolates. Furthermore, it was identified that the environmental *L. monocytogenes* isolate from company B matched those associated with the outbreak, identifying a route of transmission of the outbreak strain from company A to company B, which had purchased equipment from company A. The study demonstrated the usefulness of WGS analysis coupled with epidemiological data, in addition to making it possible to trace the spread of outbreak isolates from one food-processing facility to another.

To enhance and integrate the information obtained from the real-time WGS analysis, a multi-agency collaboration, the *Listeria* project, began in September 2013, in which the sequencing data of all the available clinical, food, and food-processing-environment isolates of *L. monocytogenes* collected in the United States were made available in a public repository [164,165]. Recently, the Centers for Disease Control and Prevention (CDC) [122] reported an outbreak of multistate listeriosis in the United States that is still under investigation, in which sixteen cases, twelve hospitalizations, and one death have already been reported. The information collected so far indicates that sliced meats and cheeses purchased from deli counters in several states were the likely sources of the outbreak. The PulseNet system is used by investigative agencies to identify possible cases of disease related to the outbreak. The WGS analyses carried out so far have shown that the bacteria isolated from the patients have a close genetic relationship [122]. However, a more in-depth

study needs to be performed, including targeted retail sampling, in order to clarify the root cause of this outbreak.

There are also reports of an ongoing listeriosis outbreak in the United Kingdom [193]. Although the strain of the pathogen associated with the outbreak has been found in other food products and environmental samples, a warning was issued to the public by the Food Standards Agency (FSA) and the UK Health Security Agency (UKHSA) not to consume Baronet semi-soft cheeses, which were recalled as they were found to be contaminated with worrying levels of *Listeria*. So far, three cases and one death have been linked, through the WGS, to the outbreak.

## 7. Prevention, Monitoring, and Control of *Listeria monocytogenes* in the Dairy Industry

The gravity of listeriosis has raised concerns among health authorities and the food industry, including the dairy sector, over the transmission of *L. monocytogenes* through the food chain. However, there is no uniform regulatory policy for the presence of this pathogen in various foods. While some countries, such as the United States, implement a "zero-tolerance" (0/25 g) policy, treating all RTE foods contaminated with *L. monocytogenes* equally [194], there is no global consensus on the matter. In contrast, other countries, such as Canada, have implemented regulatory measures based on the potential of the food to support *L. monocytogenes* growth (categorized as high- or low-risk foods) and the populations that are likely to consume them, especially susceptible individuals [195]. For high-risk foods and products intended for higher-risk groups, a "zero tolerance" *Listeria* policy is also enforced, in which *L. monocytogenes* should not be detected in 25 g of a product when evaluating one to ten food samples, depending on the country [196–199].

For RTE foods that are not conducive to the growth of *L. monocytogenes* during their shelf lives, such as foods with a pH lower than 4.4 or a water activity lower than 0.92, the presence of low levels of the pathogen ($<10^2$ CFU/g) is considered safe. However, it should be noted that different countries have varying regulatory policies regarding the testing of *L. monocytogenes* in RTE foods. Some countries, such as Brazil, do not require testing for the pathogen in foods that do not support its growth [200]. However, it is important to acknowledge that even with these policies, there is no guarantee that listeriosis will not occur, as noted by Farber et al. [201].

Due to the inherent characteristics of *L. monocytogenes* and the multiplicity of possible entry routes for the bacterium in food-processing plants, its presence in these environments is highly likely, with recalls and outbreaks of listeriosis often traced back to sources of contamination at these sites [8,94,196]. In addition to the requirement to adopt good food-manufacturing practices, industry and regulatory agencies have highlighted the need to develop and implement risk-based hygienic zoning to minimize the likelihood of transient strains of *L. monocytogenes* entering sensitive areas in the food-processing line [95]. Hygienic zoning should be designed individually for each facility, based on its production practices, the variety of its equipment, the complexity of its processing lines, and its prior environmental history of pathogens [94,196]. Furthermore, it is crucial for food industries to design and implement effective, risk-based environmental monitoring programs (EMPs) to check the effectiveness of hygienic zoning, in addition to determining whether cleaning and sanitizing procedures are carried out appropriately, as well as identifying possible shelters for the microorganism, so that corrective actions can be applied to eliminate persistent strains (a "seek-and-destroy" plan) [94,197,201]. Although this is a difficult and complex task, numerous publications and guidance documents have been developed over the last few decades describing procedures that may be effective for monitoring and controlling *L. monocytogenes* in food-producing facilities in order to reduce the frequency and level (CFU/g or cm²) of product contamination [194,195,197–199,202–204]. Many of these recommendations are applicable to environments that process dairy products, as described below.

As the presence of the pathogen is more likely in uncontrolled or raw manufacturing areas than in areas where production is controlled, one of the first important steps to prevent food contamination with L. monocytogenes is to restrict its entry into the processing environment. The industry must establish the microbiological parameters of incoming raw materials, carry out audits with suppliers, verify the conditions of the transport and reception of materials (e.g., raw-milk transport temperatures), define the conditions of access to the industry, establish sanitary conditions and production flows, and implement pest-control programs to reduce the potential for cross-contamination [52,202,205,206].

Furthermore, employees must be continuously trained in receiving materials, cleaning and sanitizing hands and equipment, the proper handling of food and packaging at all stages of processing, how to avoid cross-contamination through access control, transit- and sanitary-production flow, and facility-specific practices in general [198,201,202]. In addition, employees must be aware of the consequences of the presence of *L. monocytogenes* in food, especially with regard to the safety of individuals belonging to higher-risk groups [202,206].

The proper sanitary design of equipment and facilities is also crucial to pathogen control. Food-processing lines must have layouts that prevent cross-contamination, allow adequate water drainage, and are easy to clean and sanitize, in order to prevent the formation of niches where *L. monocytogenes* can grow and settle, forming biofilms [198,202,207]. The inspection and maintenance of floors, ceilings, drains, walls, windows, air ducts, and exhaust fans are essential to prevent the entry and persistence of *Listeria* within the industry [94,202,208]. Equipment must also be designed to both meet manufacturing needs and allow proper cleaning and sanitizing [198,202]. In addition, equipment must be regularly monitored and calibrated to ensure that processing, from raw materials to final products, is carried out appropriately [202,209].

The EMP should also contain validated specifications on cleaning and sanitization processes for both processing environments and of equipment, to prevent the contamination of food. These procedures are intended to remove organic residues and reduce or eliminate undesirable microorganisms from both food-contact surfaces and the processing environment [210]. Thus, effective chemicals, in proper concentrations, should be used by trained staff, in order to meet the requirements and the necessary frequency determined in the EMP [203,206].

A successful EMP also includes regular environmental and equipment microbiological testing for *Listeria* sp. to verify whether the control strategies in place are effective. The presence of other *Listeria* species in a sample indicates that conditions are favorable for the growth of *L. monocytogenes* and implies the need for corrective actions [201,202]. According to Gupta and Adkhari [94], a good environmental-monitoring strategy is to divide the facility into four main zones: (1) food-contact surfaces (machinery, equipment, devices, etc.); (2) areas immediately adjacent to food-contact surfaces ("indirect contact areas," such as bearings and aprons); (3) non-food-contact surfaces located within the production area (floors, drains, etc.); and (4) the non-productive areas of the facility (hallways, cafeterias, etc.). The main target areas for environmental sampling are zones 1 and 2, as they present a higher risk of the presence of *L. monocytogenes* [202].

Additionally, in order to prevent or hinder the growth of *L. monocytogenes*, the design of food products should take into account both intrinsic and extrinsic factors. It is crucial to carefully consider these factors, as even a minor lapse in the implemented preventive measures can result in food contamination [8,94,95].

Overall, implementing the strategies outlined above can significantly reduce the presence of *L. monocytogenes* in food-processing facilities, particularly in high-risk foods. This will lead to improvements in the microbiological food safety of products and, ultimately, benefit consumers. For a more comprehensive understanding of regulatory policies and the environmental monitoring of *L. monocytogenes* in food-processing environments, we recommend consulting the references cited in this discussion.

*Strategies for the Control of Listeria monocytogenes*

The control and elimination of listerial biofilms is a major problem for the food industries. Studies evaluating the effectiveness of widely used disinfectants, such as sodium hypochlorite in varying concentrations, hydrogen peroxide 2% (*v/v*), and benzalkonium chloride 200 ppm (*w/v*), have shown that these compounds were not able to fully remove listerial biofilms established on different surfaces [211,212]. The dairy industry is also confronted with the problem of the presence of *L. monocytogenes* existing in a state in dairy products and processing environments following exposure to various sublethal stresses, either independently or in combination. These stresses include unfavorable temperature, oxidative stress, and high or low osmotic stress, among others [49,213]. Bacteria in the VBNC state lose the ability to grow in conventional culture media, but can remain virulent, regain their cultivability when they encounter suitable conditions, and resist antimicrobial and sanitizing agents used in the dairy industry due to their reduced metabolic activity and altered cell morphology [49,213,214]. Thus, the VBNC state can contribute to *L. monocytogenes*' adaptation, persistence, and transmission in the dairy industry [49].

Non-thermal technologies have been explored as alternatives to heat treatment for controlling *L. monocytogenes* in milk and related products. Promising results were observed in studies that examined the efficacy of high-pressure processing (HPP), pulsed electric fields (PEFs), UV light within the range of 10–400 nm, and ultrasound [215–221]. In addition to ensuring food safety, the utilization of these technologies is also intended to minimize degradation and undesirable changes in the nutritional and sensory characteristics of products that result from heat treatment [104,222]. An additional significant benefit of some of these technologies is their applicability to packaged goods, which can help to address the issue of recontamination during processing, particularly for RTE foods [217,218]. Nevertheless, their large-scale implementation is costly, their process controls may be inadequate, regulatory approval is lacking, and there is the possibility of damaged microbial cells recovering and resuming growth [104,222].

Other approaches for inactivating biofilms, as well as persistent and VBNC cells of *L. monocytogenes* include the utilization of natural compounds, such as products of plant origin, microorganisms and their metabolites, especially lactic-acid bacteria (LAB), probiotics, postbiotics, bacteriocins, and bacteriophages, in an environmentally friendly way [214,223–227]. Although most of the studies on these approaches have been conducted in vitro, on specific foods and packaging, the products examined may eventually be applied to sanitizers and equipment in the future.

Plant-derived compounds, such as cinnamaldehyde, eugenol, resveratrol, and thymoquinone, demonstrated antilisterial activity in vitro, with variable minimum inhibitory concentrations (MICs) [228,229]. Furthermore, some of these compounds partially inhibited biofilm formation by different strains of *L. monocytogenes* on polystyrene surfaces, in a dose-dependent manner. However, studies are still needed to assess their impact on the formation of VBNC cells. Other studies used plant-derived products, such as phytochemicals (e.g., rosmarinic, ellagic, gallic, chebulinic, myristic, and chebulgic acids) and essential oils (e.g., thyme, clove, and cinnamon) to demonstrate that, using concentrations and carriers appropriate, these compounds can significantly inhibit biofilm formation or the growth of *L. monocytogenes* in dairy products, such as milk and fresh cheeses [230–232]. Nevertheless, it is crucial to conduct studies on the toxicities of these types of compounds, as well as their impact on the sensory attributes of dairy products, to determine consumer acceptability.

The use of biopreservation has been highlighted as another very interesting green strategy to improve the food safety of various food products. Biopreservation employs the accompanying microbiota of the food, particularly LAB or their metabolites, to control the development of undesirable microorganisms [233]. Lactic-acid bacteria are generally recognized as safe (GRAS) and, as well as being part of the natural microbiota of various foods, and are often used as starter cultures. Many LAB present antilisterial activity, such as *Lactococcus*, *Leuconostoc*, *Pediococcus*, and *Lactobacillus* species (now reclassified into 25 new

genera) demonstrating success in inhibiting *L. monocytogenes* and other pathogens in dairy products, such as milk, cheese, and yogurt [109,233,234]. Additionally, LAB are capable of producing bacteriocins, or bacteriocin-like inhibitory substances, which are antimicrobial peptides with inhibitory and bactericidal activities, which have been extensively studied for the biocontrol of *L. monocytogenes* in foods [235]. Although nisin, pediocin PA-1, and micocin are currently the only bacteriocins approved by the U.S. Food and Drug Administration (FDA) for use as food preservatives to inhibit undesirable microorganisms, studies have shown that several bacteriocins, not only from LAB, are effective in inhibiting *L. monocytogenes* [224,233,235]. *Bacillus velezensis* was found to strongly inhibit *L. monocytogenes* due to the production of a bacteriocin with great stability at different temperatures and pH [224]. Lactocin AL705, produced by *Lactobacillus curvatus* CRL1579, inhibited biofilm formation by *L. monocytogenes* without affecting planktonic growth [225]. Enterocin synthesized by *Enterococcus avium* DSMZ17511 was applied as a coating on different cheeses with added *L. monocytogenes*, resulting in the reduced viability of the pathogen, with a faster diffusion rate in softer and higher-moisture cheeses [226].

Another major use of the incorporation of desirable microorganisms into food is that many of them have probiotic characteristics, in addition to improving the sensory characteristics and safety of food products. Using probiotics and their postbiotic metabolic by-products, which have GRAS status and are already used in food and medicine, is a promising strategy to inhibit *L. monocytogenes* in dairy food. Ewida et al. [118] conducted a study in which soft cheeses were made with pasteurized milk, *L. monocytogenes*, and probiotics isolated or in combination (*Bifidobacterium breve* and *Bifidobacterium animalis*). The addition of *B. animalis* or the combined species reduced *L. monocytogenes* during the 28-day ripening process. Moradi et al. [223] assessed postbiotics from *Lactobacillus acidophilus*, *Lactobacillus casei*, and *Lactobacillus salivarius* in vitro and in pasteurized whole milk against *L. monocytogenes* and its biofilm. The postbiotics from all the *Lactobacillus* species completely or partially inhibited *L. monocytogenes* growth in a pH range of 4 to 9. This antibacterial activity was associated with pyrrolo [1,2-a] pyrazine-1,4-dione and, to some extent, with different organic acids. These postbiotics also demonstrated the ability to remove in vitro biofilms formed by *L. monocytogenes*, depending on the type of postbiotic and the duration of contact. Additionally, in pasteurized whole milk refrigerated at 4 °C for three days, the postbiotics showed antilisterial activity.

The use of lytic bacteriophages has also been successfully tested for the biocontrol of *L. monocytogenes* in different food matrices, such as artificially contaminated vegetables and dairy products [227,236]. Lytic bacteriophages are viruses that hijack the metabolic mechanisms of their hosts for their growth and multiplication, leading the death of host cells due to lysis, without disturbing the normal microbiota of the food, as there is a high specificity of phage-host interactions [236]. Lee et al. [237] showed that different lytic bacteriophages inhibited the growth of *L. monocytogenes* in a culture medium and milk at various temperatures, while Silva et al. [238] observed that the application of a lytic phage in soft cheeses reduced the initial pathogen counts only at the beginning of the cold storage of the products, indicating that the effectiveness of the phage treatment was dependent on the initial contamination of the product with *L. monocytogenes*. Guenther and Loessner [239] used a lytic bacteriophage to control *L. monocytogenes* during the production and ripening of soft-ripened cheeses, and observed that a single larger dose of the phage was more effective than repeated smaller applications at reducing the pathogen counts. In addition, bacteriophages were able to reduce *L. monocytogenes* counts below the detection limit when low initial counts of the pathogen were added to cheeses, which was accordance with Silva et al. [238]'s observations. Although the use of phages for the biocontrol of *L. monocytogenes* may offer a safe and ecological approach to reducing the contamination of foods with the pathogen, further studies are needed to optimize their use in different food matrixes and environments.

Combined strategies can also be employed in order to control *L. monocytogenes* in the dairy industry. Soni et al. [240] combined the effect of three GRAS antimicrobials

(bacteriophage P100, lauric alginate, and a potassium lactate–sodium diacetate mixture) to inhibit *L. monocytogenes* in fresh cheese and observed both listericidal and listeriostatic effects. Similarly, Komora et al. [241] observed a synergistic effect of the combination of bacteriophage P100, bacteriocin pediocin PA-1, and mild–high hydrostatic pressure in the reduction in *L. monocytogenes* counts in UHT milk.

While numerous studies have investigated strategies to inhibit *L. monocytogenes* and eradicate its biofilm, most of these methods have not been implemented on an industrial scale. Further research is needed to confirm their effectiveness in large-scale production. In the dairy industry, it is important to identify critical control points to plan and update *L.-monocytogenes*-management plans in food-safety-management programs. Policy makers can also use data on the occurrence of this pathogen and outbreaks of listeriosis in milk and dairy products to provide mandatory notification of listeriosis outbreaks, establish microbiological criteria, plan inspections and analyses, and develop educational campaigns for health professionals and high-risk groups [193,201].

## 8. Conclusions and Perspectives

Significant knowledge has been gained about the physiological and genetic mechanisms employed by *L. monocytogenes* to persist in industrial settings, as well as to cause multiple types of manifestation in its hosts. The ways in which this bacterium can circumvent the body's defense systems have been extensively studied, revealing sophisticated mechanisms of infection and an orchestrated use of different virulence factors that protect the bacterial cell from the host's immune system. Additionally, the ability of *L. monocytogenes* to survive or even grow in harsh environments, including refrigeration temperatures, makes this pathogen a special concern for the dairy industry. Food-management systems that employ hazard analysis and critical control point (HACCP) principles, robust prerequisite programs focused on the hygienic design of equipment, sanitation, and product formulation, and proactive environmental monitoring programs are important ways of reducing the likelihood of the installation of *L. monocytogenes* in the food-processing plant. The development of molecular technologies to detect the spread of the pathogen, as well as to elucidate outbreaks of listeriosis, has certainly been a substantial gain.

However, there are still gaps that need to be filled with regard to pathogen control in food-processing and -marketing environments, as outbreaks of listeriosis in dairy products continue to be identified. Thus, according to the findings of this review, it is clear that additional measures are required. These include the following: the creation and widespread dissemination of more educational materials to inform the public about the dangers of consuming dairy products made from raw milk; the adoption of mandatory notification of listeriosis outbreaks; the establishment of more rigorous policies for monitoring and controlling *L. monocytogenes* in processing environments; an increase in the frequency of inspections of dairy products and facilities; the implementation of quality programs in dairy industries with employee training; and the development of new methods for controlling *L. monocytogenes*.

It is important to note that there may be limitations in this review in relation to the selection of the articles, which could have introduced some bias.

**Author Contributions:** Conceptualization, V.F.A., U.M.P. and F.A.A., writing—original draft preparation, A.C.R., F.A.A., M.M.M. and B.R.M.; writing—review and editing U.M.P. and V.F.A.; project administration, V.F.A.; funding acquisition, U.M.P. All authors have read and agreed to the published version of the manuscript.

**Funding:** This research received no external funding.

**Institutional Review Board Statement:** Not applicable.

**Informed Consent Statement:** Not applicable.

**Data Availability Statement:** As this is a comprehensive review, this manuscript was based on published literature reports, which were all properly referenced.

**Acknowledgments:** U.M.P. acknowledges the São Paulo Research Foundation for financial support (grant 2013/07914-8) to the Food Research Center. A.C.R. acknowledges the São Paulo Research Foundation for a technical training scholarship (grant 2022/12535-5).

**Conflicts of Interest:** The authors declare no conflict of interest.

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
