# Peer review of "Listeria monocytogenes: An Inconvenient Hurdle for the Dairy Industry"

_2624-862X, doi:10.3390/dairy4020022_

Round 1

Reviewer 1 Report

The abstract contains main required components and it forms coherent text with logical conclusions and interactions between its immanent parts. The title of the paper is well formulated and it covers the content. The introduction logically follows the aim of the paper and it provides valuable introspection into a timely topic of interest for Dairy’s readership such as the occurrence of listeria monocytogenes in the dairy industry. Methodological part of the paper is suits current scientific standards. Results are presented clearly. The interpretation of tables is acceptable. The level of the author’s knowledge is satisfying. It is obvious that authors are well oriented in the topic and that they use appropriate terms. The overall level of language is appropriate. However, a strongly recommend the authors to:

-        Expand the discussion on section 7 “Prevention, monitoring, and control of Listeria monocytogenes in the dairy industry” by adding discussion on how industries and policymakers may contribute to low the occurrence of listeria detection in the food supply chain;

-        A large part of the conclusions is not appropriate, as it is only a concise repetition of the comments. In the conclusions, you should simultaneously consider all you have discovered, and exploit it to add something new (or new interpretations), and policy indications.

-        The authors do not discuss possible limitations of their study or the insights for future directions of research.

-        I recommend that authors review the article thoroughly and consider using a professional proofreading service to improve the style of the article. Many sentences are unclear.

-        Tables’ readability has to be improved.

Author Response

We are immensely grateful for the enriching considerations of the reviewer.

 1)  Expand the discussion on section 7 “Prevention, monitoring, and control of Listeria monocytogenes in the dairy industry” by adding discussion on how industries and policymakers may contribute to low the occurrence of listeria detection in the food supply chain;

Response:  We redid the item 7 as requested, following the reviewer's guidelines, and we hope it is clearer and more objective now. In addition, we also did a complete review of item 7.1, as we thought this revision would facilitate reading the manuscript and  improve its quality.

2)   A large part of the conclusions is not appropriate, as it is only a concise repetition of the comments. In the conclusions, you should simultaneously consider all you have discovered, and exploit it to add something new (or new interpretations), and policy indications.

The authors do not discuss possible limitations of their study or the insights for future directions of research.

Response: Thanks again for the reviewer comments. As the subject is quite broad, we  tried to make a short overview of the work in the final conclusions, inserting new considerations, including the main limiting factor of the study, as requested.

3) I recommend that authors review the article thoroughly and consider using a professional proofreading service to improve the style of the article. Many sentences are unclear.

Response: The manuscript was carefully read by a native speaker and modifications were made, as per his guidelines, to clarify the sentences.

4) Tables’ readability has to be improved...

Response:  All tables have been revised and we have tried to organize them in a way that should be easier to understand.

Reviewer 2 Report

I read the manuscript carefully. I believe that the authors have done a great job and that in one place you can find a lot of information about the pathogen and its role as a biological agent in the dairy industry. The work contains the necessary chapters and sub-chapters that make the reader follow the text.

The authors very clearly presented all the relevant data related to the bacterium itself, its pathogenesis, evidence procedures, ways of transmission and disease-causing in humans, and finally the possibilities we have in preventing its occurrence in dairy industry facilities and in products.

The tables are clear and interesting and provide an insight into the presence of the bacteria and substantiate its danger.

Author Response

We are immensely grateful to the reviewer for his considerations and we are glad that you liked our manuscript.

Round 2

Reviewer 1 Report

The authors properly addressed my comments. I am fine with the current version.